# Evaluating Snow in EURO-CORDEX Regional Climate Models with Observations for the European Alps: Biases and Their Relationship to Orography, Temperature, and Precipitation Mismatches

**Michael Matiu** *[ID], **Marcello Petitta, Claudia Notarnicola**[ID] **and Marc Zebisch**

Institute for Earth Observation, Eurac Research, 39100 Bolzano, Italy; marcello.petitta@eurac.edu (M.P.); claudia.notarnicola@eurac.edu (C.N.); marc.zebisch@eurac.edu (M.Z.)
* Correspondence: michael.matiu@eurac.edu

**Abstract:** Climate models are important tools to assess current and future climate. While they have been extensively used for studying temperature and precipitation, only recently regional climate models (RCMs) arrived at horizontal resolutions that allow studies of snow in complex mountain terrain. Here, we present an evaluation of the snow variables in the World Climate Research Program Coordinated Regional Downscaling Experiment (EURO-CORDEX) RCMs with gridded observations of snow cover (from MODIS remote sensing) and temperature and precipitation (E-OBS), as well as with point (station) observations of snow depth and temperature for the European Alps. Large scale snow cover dynamics were reproduced well with some over- and under-estimations depending on month and RCM. The orography, temperature, and precipitation mismatches could on average explain 31% of the variability in snow cover bias across grid-cells, and even more than 50% in the winter period November–April. Biases in average monthly snow depth were remarkably low for reanalysis driven RCMs (<approx. 30 cm), and large for the GCM driven ones (up to 200 cm), when averaged over all stations within 400 m of altitude difference with RCM orography. Some RCMs indicated low snow cover biases and at the same time high snow depth biases, and vice versa. In summary, RCMs showed good skills in reproducing alpine snow cover conditions with regard to their limited horizontal resolution. Detected shortcomings in the models depended on the considered snow variable, season and individual RCM.

**Keywords:** climate change; MODIS snow cover; station snow depth; Alps; mountain climate

## 1. Introduction

In mountain climates, snow is a systemic component besides temperature and precipitation. While future changes in temperature and precipitation have been assessed using climate models, little has yet been done for snow with climate models, most likely because general circulation models (GCMs) have a too coarse resolution (>100 km) and regional climate models (RCMs) have only recently reached reasonable horizontal resolutions (10–50 km) to study changes in snow parameters. This provides an alternative to the traditional approach of using hydrological or dedicated snow models, which are forced by temperature and precipitation from climate models [1].

The horizontal resolution of RCMs is now widely in the range of 0.44° (~50 km) to 0.11° (~12.5 km), see e.g., the CORDEX initiative For Europe, most simulations are also available at the higher resolution, via the World Climate Research Program Coordinated Regional Downscaling Experiment (EURO-CORDEX). For complex mountainous terrain such as the European Alps, this resolution is moderate, however, the RCMs cover the whole domain and thus have a good trade-off between

resolution and coverage. It should be noted that RCMs have been run also at higher resolutions (2.2 km with a convection permitting scheme [2]) and employed for snow cover studies [3]. Due to high computational requirements, such simulations are still rare (i.e., not available as model ensemble) and do not cover longer temporal ranges (e.g., like the full 21st century).

Large-scale and high-resolution validation of snow in RCMs is hampered, because there are only few observational data sets, especially for snow water equivalent (SWE). Nevertheless, there have been regional attempts. Salzmann and Mearns compared multiple RCMs at 50 km resolution with station observations and reanalysis in the Upper Colorado River basin in the USA and found that RCMs simulated too little SWE with a too-late start and too-early end of snow cover, most likely because of unresolved topography and model schemes [4]. Steger et al. compared RCMs at 25 km with a gridded SWE product for Switzerland and found that, in general, RCMs reproduced spatial and seasonal variability but underestimated SWE at low elevation and overestimated SWE at high elevations [5]. Da Ronco et al. compared snow cover in the RCM COSMO-CLM with remote sensing observations from MODIS for the Po river basin in Italy and found a good agreement regarding duration, extent, and interannual variability, but a too-early snow melt in the RCM. Additionally, they also highlighted the importance of horizontal resolution [6]. Terzago et al. provide the most comprehensive evaluation of SWE in RCMs (and GCMs) so far using multiple remote sensing products and reanalysis for the entire European Alps and found that RCMs overestimated SWE and only ERA-Interim driven RCMs had a comparable amplitude in the snow cycle while GCM-driven RCMs had large positive biases [7].

In this study, we evaluate snow in the 0.11° (~12.5 km) EURO-CORDEX RCMs using high-resolution observations. This is the first time, to the best of our knowledge, that a high-resolution comparison has been performed using such an extensive set of RCMs and covering the whole Alpine domain. Going to high resolution allows to study the effects of the orographic misrepresentation [8] and temperature biases [9] explicitly. Specifically, we use snow data from the EURO-CORDEX RCMs, and perform an inter-comparison of snow variables in RCMs, and evaluations of snow cover with remote sensing (MODIS) and of snow depth with stations. Both evaluations consider relationships to orography mismatch and to temperature biases (E-OBS and station). Precipitation biases are only considered with snow cover and E-OBS. The aim is to evaluate the suitability of modelled snow in RCMs, and to quantify the magnitude of the biases as well as the extent to which they can be related to the erroneous representation of orography as well as temperature and precipitation forcing.

The paper is divided as follow: Section 2 is devoted to describing the data and the methodology used, Section 3 presents the results of the comparisons of the models with the observations and discusses them, and Section 4 summarizes the conclusions and the outlook of the work.

## 2. Materials and Methods

### 2.1. Data

#### 2.1.1. Climate Models (RCMs) Snow and Temperature

The climate model data originated from the EURO-CORDEX initiative, whose data is available via the ESGF (Earth System Grid Federation) servers. We used all RCMs that provide at least one of the following snow variables, both from reanalysis and GCM driven runs: snow cover fraction (SNC; unitless; originally with range 0–100 and converted to fractions 0–1; corresponds to the fractional area of the grid cell that is snow covered), snow depth (SND; provided in meters), or snow amount (SNW; water amount in snow, usually also named SWE—snow water equivalent; provided in kg m$^{-2}$). Not all modelling groups supply all snow variables. Table 1 provides an overview of which variables were available for which RCM. For brevity, from now on, only the RCM name and not the modelling institute shall be used.

For the GCM driven runs, the historical period ends in 2005, and then the different RCP (representative concentration pathway) scenarios start. The gridded remote sensing snow cover data set covers the period 2002 to 2019, so we merged the historical run with the RCP8.5 scenario to

maximize the temporal overlap. This was done if the historical and scenario run came from the same ensemble and downscale realization. RCP8.5 (the high emission scenario) was chosen over the other scenarios (RCP2.6, RCP4.5, RCP6.0), because the middle scenarios (RCP4.5 and RCP6.0) had only few model runs thus severely limiting data availability, and RCP8.5 represented the actual emissions after 2006 better than RCP2.6 (see [10] for an assessment until 2012). The complete list of RCMs used can be found in Table S1.

**Table 1.** Overview of regional climate models (RCMs) and snow variables (SNC: snow cover fraction; SND: snow depth; SNW: snow amount) that were available in this study. Meaning of cell content: empty = variable not available from ESGF (Earth System Grid Federation) servers; X = variable available for both reanalysis and general circulation models (GCM) driven runs; G = variable only available for GCM driven runs; (G) = variable only available for specific GCM driven runs.

| Modelling Institute | RCM | SNC | SND | SNW |
|---|---|---|---|---|
| CNRM | ALADIN53 | | | X |
| CNRM | ALADIN63 | G | G | G |
| CLMcom | CCLM4-8-17 | X | X | X |
| DMI | HIRHAM5 [1] | G | G | G |
| KNMI | RACMO22E | X | X | X |
| SMHI | RCA4 | X | X | (G) |
| ICTP | RegCM4-6 | | | X |
| MPI-CSC [2] | REMO2009 | | | X |
| GERICS | REMO2015 | | | X |
| IPSL-INERIS | WRF331F | X | X | |
| IPSL | WRF381P | X | | X |

[1] HIRHAM5 actually also has snow variables also for the evaluation runs, but only for downscale realization v1, and in this study only v2 and v3 were used because of some errors in v1 (see also Section 2.1.1). [2] MPI-CSC is now GERICS.

Since the data have been initially published, some errors in the data were discovered (see also Errata Table [11]). Regarding snow, these are snow accumulation in RACMO22E (issue number 12 and 13 in [11]), snow accumulation and errors in SND for HIRHAM5 (issue number 18 and 33 in [11]; solved by using only versions above v2), and errors in SNW for RCA4 (issue number 29 in [11]; solved in data after 20 August 2018).

The snow accumulation issue (grid cells showing constant snow cover and accumulating unrealistic snow towers) was found true not only for the abovementioned RCMs. While RACMO22E was the most affected, also ALADIN63 and WRF331F, and partly HIRHAM5, indicated this issue (see Table S2). The maximum daily SND in the study region (Figure 1) for all historical GCM-driven runs and the evaluation run was 340 m for ALADIN63, 33 m for HIRHAM5, 464 m for RACMO22E, and 172 m for WRF331F. But, also SNW amounts were unrealistically high with maximum values between 38,000–236,000 kg m$^{-2}$ for several RCMs (ALADIN53, ALADIN63, WRF381P, RACMO22E, and RCA4). Because this issue only affects some high-altitude grid cells, we decided to exclude these grid cells based on SND and SNW thresholds. A grid cell was removed for a given RCM, if its daily SND was above 10m and/or SNW was above 7000 kg m$^{-2}$ during any historical or evaluation run. These thresholds were chosen, because for SND, the 10 m are used internally in RCA4, too, and for SNW, it corresponds to 10 m SND with a conservative high limit on snow density. RACMO22E was most affected with 223 grid cells removed (4.4% out of 5046 total land grid cells in the study area), followed by HIRHAM5 with 152 grid cells (3%), while the other RCMs had 15–71 (0.3–1.4%) affected cells (Table S3 and Figure S1).

In addition to snow variables, also 2 m temperature (TAS, Kelvin), precipitation (PR, mm s$^{-1}$) and model grid altitude (OROG, meter above sea level) were used. Temperature was converted to degrees Celsius, precipitation to mm day$^{-1}$. For WRF381P no altitude grid was available from ESGF servers, so it was removed from most of the analysis, since they required some altitude information.

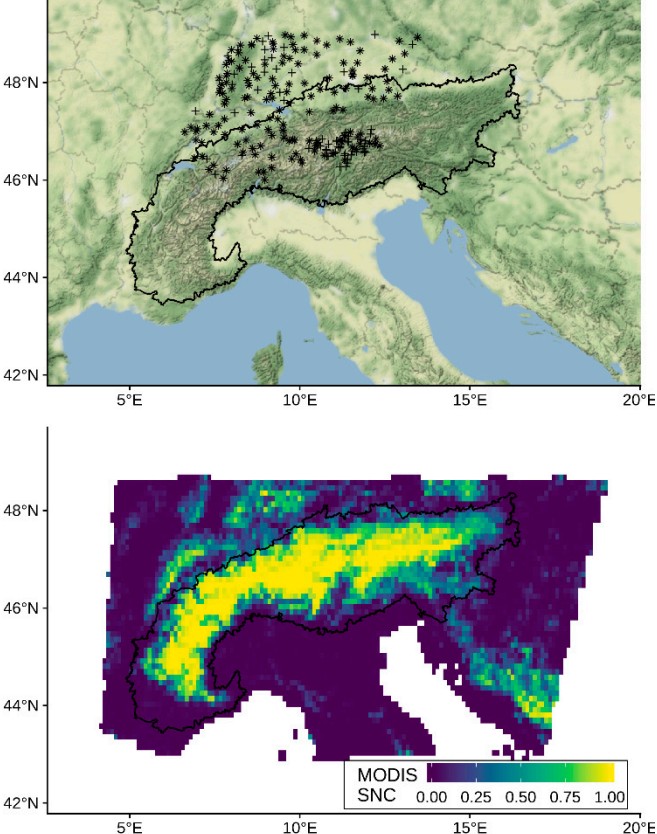

**Figure 1.** Map of study region (European Alps). (**Top**) Terrain map with boundaries of the Alpine Convention (Alpconv), marking the core Alpine region. Points indicate the station locations (+: used for reanalysis driven RCMs; x: used for GCM driven RCMs; stars (+ and x overlaid): used for both). (Map tiles by Stamen Design, under CC BY 3.0. Data by Open Street Map, under ODbL). (**Bottom**) Example map of MODIS snow cover fraction (SNC) for 1 January 2012, upscaled to RCM resolution (~0.11°).

### 2.1.2. E-OBS Temperature and Precipitation

For gridded temperature and precipitation fields, we used E-OBS v20.0e [12], which is a gridded product based on the interpolation of station data. The mean temperature and precipitation are available in daily resolution for the period 1950–2018 at 0.1° horizontal resolution for Europe. Auxiliary data consisted of the elevation grid. The data was downloaded from Copernicus servers.

### 2.1.3. Remote Sensing (MODIS) Snow Cover Fraction

The gridded observational snow product applied in this study is snow cover from remote sensing [13]. We used a MODIS-based product that has been specifically tailored to the complex terrain in the Alps [14]. It spans the period July 2002 until May 2019 at daily resolution and covers the whole alpine arc plus some surroundings at 250 m horizontal resolution (Figure 1). Clouds have been removed using temporal and spatial filters [13], in order to be able to conduct climatological comparisons.

MODIS is only available since relatively recent periods, and while there are other remote sensing products that cover more of the past such as e.g., AVHRR (Advanced Very High Resolution Radiometer) at coarser spatial resolution, we still decided to use MODIS due to the following points. First, it has a higher accuracy (even when aggregated) than the older satellites. Second, the implementation is specifically tailored to the complex terrain found in the Alps. Third, it is in daily resolution and, thanks to the cloud-filtering, has complete time series. Fourth, this study is part of a project aiming at downscaling snow cover from regional climate models, and MODIS is the only product that offers a high resolution with a long enough period.

### 2.1.4. Reference Orography

The elevation of RCM grid cells is smoothed out because of numerical stability. The E-OBS elevation is also smoothed because of the interpolation. Thus, even if the model topography is based on an accurate digital elevation model, it differs from real topography. Since one aim was to investigate the effects of this orographic mismatch, we needed a reference "true" orography. For this we took the digital elevation model used in the MODIS processing [13], which is based on the SRTM (Shuttle Radar Topography Mission), and which has been aggregated to MODIS resolution. In the following this "true" elevations will be called "MODIS elevation", implying this is the observed (and not model) topography and that it corresponds to the MODIS data.

### 2.1.5. Station Snow Depth and Temperature

Station data was collected from the Hydrological office of the province of Bolzano in Northern Italy, from the German weather service (DWD) for Germany, and from MeteoSwiss for Switzerland. Stations that fell in the study region (see top of Figure 1) and had snow depth and temperature measurements were selected. The distribution of meteorological stations is known to be not homogeneous in space nor along the altitudinal gradient. In our case, the stations sampled the central part of the study area, both north and south of the main ridge. The altitudinal distribution of the stations encompassed the distribution of the coarse grid cell altitudes (at ~12.5 km) from 0 to 3000 m; however, since many stations were located in the central part of the Alps, their relative frequency was higher between 500 and 2000 m, and lower otherwise (Figure S2). The data was selected and preprocessed in order to have high temporal coverage with minimal missing data.

For this, nearby stations were merged if their horizontal distance was less than 1 km and their altitude difference less than 100 m. Then some basic quality threshold on the snow depth data were applied (no negative values, not more than 250 cm snow fall in one day, not more than 70 cm snow melt in one day; see also [15]). Afterwards an initial subset was taken, that consisted of stations with starting year before 1990 and ending after 2000, with at least 15 years of observations, and not more than 8 missing years in any 11 year window. With this initial subset, a gap-filling was applied to reduce the number of missing observations, similar to what has been done in temperature and precipitations climatological studies [16,17]. The gap-filling is based on up to three neighboring stations, if they show a high correlation (>0.8) with the station with missing data, and if there is enough data in common (at least 150 days in an adaptive window of at most 91 days in a fixed 11 year period centered around the missing day). Additive (for temperature) and multiplicative (for snow depth) scaling factors were calculated for the neighboring stations with respect to the reference station. The imputed value is then a distance weighted average of the scaled values of the neighboring stations. Then, monthly averages were calculated if at least 90% of the daily values were available.

Finally, two separate subsets were made for reanalysis- and GCM-driven RCMs, since for reanalysis driven RCMs year-to-year correspondences can be expected, while for GCM driven RCMs only climatological comparisons are meaningful. For reanalysis driven RCMs, only the common period for all RCMs (1989–2008) was considered, and stations were selected if they had at least 10 years of data. For GCM driven RCMs, only the common period 1970–2005 was considered, and stations were selected if they had at least 26 years of data within that period. Note that the selection was done separately per month, so slightly different station numbers were available for each month. This was done, because for most stations, recording was stopped at the end of the snow season and sometimes even before. Thus, using only complete time series for all months would have drastically reduced the number of stations. Anyhow, the number of stations was almost constant across the seasons (winter, spring, . . . ). In the end, ~208 stations were available for reanalysis driven RCMs and ~130 for GCM driven RCMs (see Figure 1 for a map of the station locations).

## 2.2. Methods

### 2.2.1. Preprocessing and Spatial Alignment

Since all gridded fields were in different horizontal resolutions and projections, they had to be aligned first to make comparisons possible. Since the RCMs were the objects of interest, we decided to keep the RCM resolution as reference and take the MODIS projection as target. The RCM resolution was 0.11°, which is approximately 8.5 × 12.25 km in the European Alps. The MODIS data had a higher resolution of 250 m (231 m in projection) and was upscaled to match grid cells of approximately 8.5 × 12.25 km size by using factors 37 and 53. For each coarse resolution grid cell, this made the calculation of fractional snow cover possible using the binary snow/land MODIS data. Snow cover fraction was calculated as fraction of number of pixels of snow/(land + snow), not counting cloud pixels, and only if there were less than 30% cloud pixels. Elevation of the coarse resolution MODIS grid was taken as average elevation of all fine grid cells.

The RCM data was then projected onto the coarse resolution MODIS grid, which was in LAEA (Lambert azimuthal equal-area) projection, using CDO (climate data operators) and nearest-neighbor interpolation. By providing the reference target grid, the interpolation with CDO worked for both normal and rotated-pole grids found in RCMs. All RCM variables (SNC, SND, SNW, TAS, PR, and OROG) were projected using the same method to preserve within grid cell relationships, and nearest-neighbor was chosen over e.g., bilinear, because bilinear introduced spurious min/max values for the fractional variable snow cover. The E-OBS grids (mean temperature, precipitation and elevation) were bilinearly interpolated onto the target grid.

### 2.2.2. Comparison of Gridded Data

Having all gridded data on a common reference grid allows comparisons on a per-cell basis for (1) snow variables (SNC, SND, SNW) among different RCMs, and (2) the variables SNC (RCM vs. MODIS), TAS and PR (RCM vs. E-OBS), and elevation (RCM, MODIS, E-OBS).

To compare snow variables within RCMs, we used daily values for the years 1971 to 2000 and from only one GCM (since differences in RCMs driven with different GCMs are low) to reduce the amount of data. Comparisons with MODIS and E-OBS were done for monthly averages over the whole time period (i.e., over all years). For the reanalysis driven RCMs, the maximum overlap of all RCM periods with MODIS and E-OBS was October 2002 to September 2008. While the period does not sample a climatology, average values should still be comparable, since the lateral forcing ensures a year-to-year correspondence. For the GCM driven RCMs, the common period was October 2002 to May 2019, which is still not large in climatological terms (e.g., 30 years), but represented the maximum temporal overlap.

In order to account for the orographic mismatch, we calculated the differences between RCM and MODIS and between RCM and E-OBS. Since the temperature difference between RCM and E-OBS can also be caused by their different orography, the temperature difference has been lapse-rate adjusted for the different elevation in each RCM versus E-OBS using monthly lapse rates as given in [18].

### 2.2.3. Comparison of Gridded with Point Data

Comparing grid cell area averages to single point observations is problematic, and should be avoided if possible. However, in certain cases, such as for snow depth or snow water equivalent, there are no high-quality high-resolution gridded data sets, so comparing grid cells to station data cannot be avoided. See also [4] for the issues and needs on comparing point scale observations to area averages.

Grid cell quantities in RCMs do, due to sub-grid variability, typically not correspond well with station data, concerning both their spatial and temporal variability. These scale issues have to be kept in mind when analyzing and interpreting results. Some of the temporal small-scale issues have been accounted for, by using temporal averages. Spatial averages (e.g., per grid cell) have not been considered, because of the large heterogeneity of station distribution and the complex terrain.

Technically, the comparison between point and grid cell was made by selecting the grid cell, in which the point observation fell. For the reanalysis driven RCMs, the time series of monthly snow depth were selected to match the available years that each station and RCM had in common. These years could also have some gaps in between, but they had to be at least 10 years in common. Then, these values were averaged over all years. Contrarily, for the free-running GCM driven RCMs, selecting single common years for short periods is not appropriate. Instead monthly climatological values were calculated for the period 1970–2005, which is the common historical period for RCMs. The same was done for the stations, which had to have at least 26 years of available observations (see also Section 2.1.5.)

### 2.2.4. Statistics, Software, and Data

Linear regression models were applied to capture the contribution of altitude, temperature and precipitation differences on differences in snow cover. The model's response variable was the snow cover bias and it was estimated using four sets of explanatory variables: (1) the altitude difference, (2) the temperature difference, (3) the precipitation difference, and (4) all these three differences. We computed the percentage of explained variance adjusted for the number of explanatory variables (i.e., the adjusted R squared) for each model, which then allows to judge the importance of each variable in isolation, and its share in the model that includes all three. The models were run separately for each month, each GCM-RCM combination, and for different altitude classes (<500, 500–1000, 1500–2000, and >2000 m).

All calculations were performed in R version 3.6.1 [19], while making heavy use of the data.table [20] and ggplot2 [21] packages. The code has been deposited at [22].

## 3. Results and Discussion

### 3.1. Snow Variables in RCMs

First, some context shall be provided by showing the relationships between the three snow variables in RCMs, and how these relationships differ between RCMs. The closest link was between snow depth (SND) and snow amounts (SNW) for the five RCMs that provided both variables. The relationship was perfectly linear for HIRHAM5 (slope set to 300 kg m$^{-3}$; see also Issue 33 in [11]), almost perfectly linear for CCLM-8-4-17 and RACMO22E. ALADIN63 and RCA4 revealed more distinct variations in snow density (Figure S3). Slopes of linear models of SNW by SND were between 289 and 386 kg m$^{-3}$.

Six RCMs allowed a comparison between SND and SNC (Figure 2). SNC is strongly linked to SND with SNC increasing along with SND and saturating at some point. However, the strength of the link and the saturation point depended on the RCM. For CCLM-8-4-17 and RACMO22E, most of the SNC variability was up to ~20 cm of SND, and SNC was 100% for almost all values of SND above 30 cm. For WRF331F, SNC showed more variability up to ~100cm of SND. For ALADIN63 and RCA4, the SNC was not so tightly linked to SND as for the other RCMs. Many (high-altitude) points showed high SND values (>2 m) with only partial SNC, while for the lower altitudes SNC was 100% also for lower SND. The SNC values of HIRHAM5 were different from all other RCMs: None showed a full snow cover (SNC = 100%), but most of the SNC values were below ~30% with corresponding SND values up to 8 m.

SNC is typically parametrized by SNW and not SND; nevertheless, we showed the relationship to SND, because later SND is compared to observations and not SNW. Moreover, since SND and SNW were highly correlated, similar results were observed when comparing SNW and SNC (Figure S4).

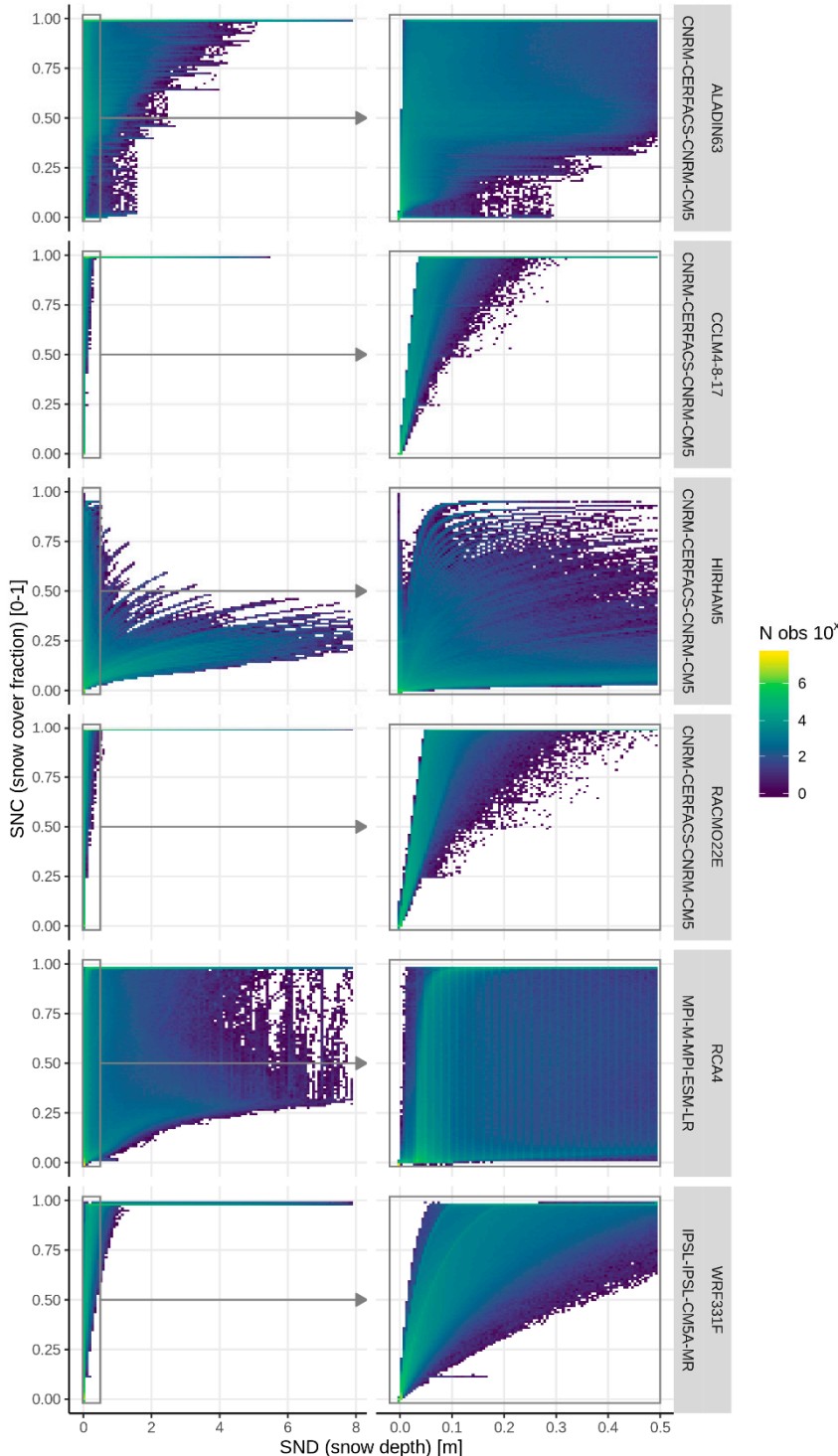

**Figure 2.** Daily SNC versus SND for the period 1971–2000 and all grid cells in the study region by RCM. Two-dimensional histograms are shown with colors referring to the number of grid cells (in log10). The GCM is denoted in the row labels on the right. Left panels show the entire SND range, while right panels present a magnification of the 0–50 cm SND range.

### 3.2. Large Scale Comparison of Snow Cover in RCMs to MODIS

The annual cycle of alpine-wide averages of snow cover was represented in reanalysis driven RCMs (Figure 3), as well as in GCM driven RCMs (Figure S5). Except for RACMO22E, which consistently overestimated snow cover, and HIRHAM5, which consistently underestimated snow

cover, all RCMs reproduced the seasonal dynamics remarkably well. Reanalysis driven WRF331F showed the best overlap with the inner higher altitude Alps (inside the alpine convention boundaries, see also Figure 1), while CCLM4-8-17 and RCA4 slightly overestimated winter snow cover (Figure 3). In the lower altitude surroundings of the Alps (outside the alpine convention boundary), all RCMs tended to underestimate snow cover, except for RACMO22E. GCM driven RCMs showed a similar pattern (Figure S5) with respect to the different RCMs, while differences in RCMs forced with different GCMs were small.

These differences in snow cover could partly be related to the seasonal cycle of the temperature bias (Figure 3 and Figure S5). For instance, WRF331F, which showed the best overlap for snow cover, also had the lowest temperature bias in winter (actually, almost none), while CCLM4-8-17 had an average temperature winter bias between −1 and −2 °C. However, the temperature bias could not explain everything, as e.g., RCA4 and RACMO22E had similar temperature biases up to −4 °C, but their snow cover fraction differed up to ~20% in absolute terms.

Precipitation biases were in the range of 20–80 mm for all RCMs inside the alpine convention boundaries, except for summer. However, no relationship was detected to the snow cover biases at this large scale: RACMO22E, for instance, had the lowest precipitation biases but the highest snow cover biases.

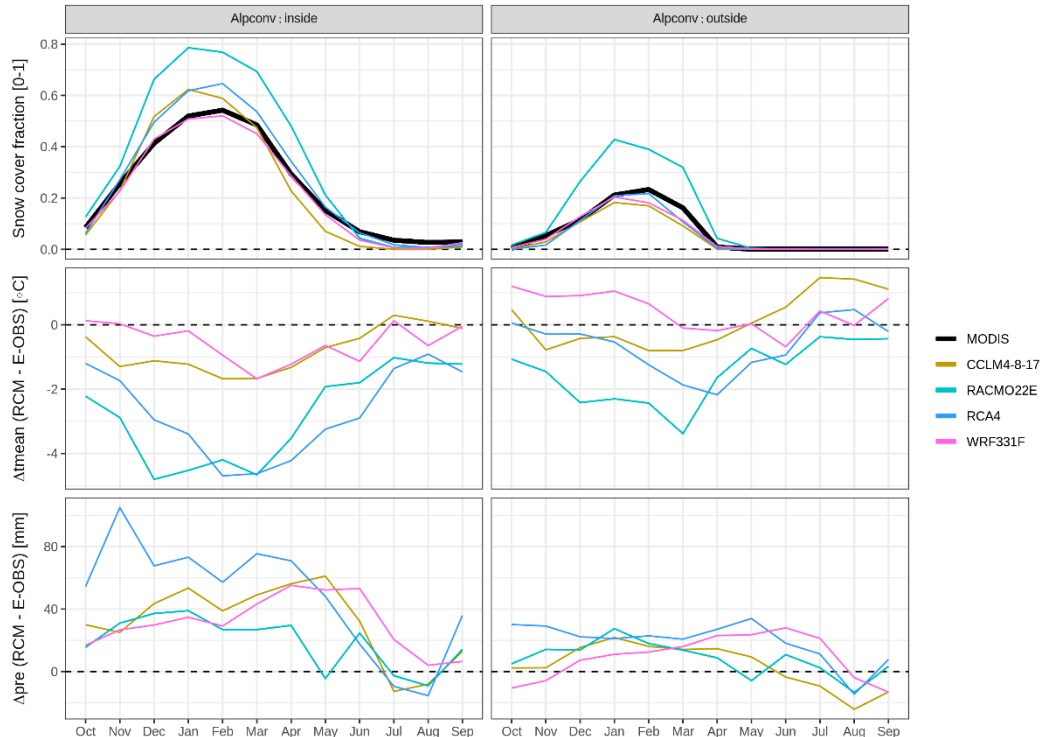

**Figure 3.** Comparison of reanalysis driven RCMs with observations (MODIS for snow cover, E-OBS for temperature and precipitation). Monthly averages are shown for inner (inside Alpconv) and outer (outside Alpconv) Alps for the period October 2002 to September 2008 for reanalysis driven RCMs (see Figure 1 for the Alpconv boundaries). Top panels show snow cover fraction over the year, middle panels show the difference between RCM and E-OBS mean temperature, and bottom panels show the difference between RCM and E-OBS precipitation.

The orographic smoothing of RCMs also could not explain the snow cover biases, since the distribution of the difference between RCM and MODIS orography was symmetric around zero (Figure S6). However, the altitude differences were quite large. For the higher altitude region (inside the alpine convention boundary), half of the grid cells differed in their altitude by more than 80–130 m depending on RCM, and 10% of the grid cells differed even by more than 280–400 m. For the lower

altitude region (outside the alpine convention boundary), 90% of the grid cells differed by less than 100–170 m.

No overarching statements can be made for all RCMs, but the biases depended on RCM, month, high vs. low altitude and whether they were driven by reanalysis or GCMs. For example, reanalysis driven WRF331F reproduced the seasonality and the amplitude of high-altitude Alpine snow cover almost perfectly, however it underestimated the low-altitude snow cover. The GCM driven version had in both cases more snow cover. Otherwise, high-altitude winter snow cover was overestimated by RACMO22E, CCLM4-8-17, RCA4, and ALADIN63. On the other hand, low-altitude early spring snow cover was underestimated by CCLM4-8-17, reanalysis driven WRF331F, and GCM driven RCA4. RACMO22E strongly overestimated snow cover in all months and all altitudes, while HIRHAM5 had the lowest snow cover extent, far away from the observed and the other RCMs, which was likely related to a rather strange snow cover parametrization (see Figure 2 and Figure S4).

This overestimation for high and underestimation for low altitudes has been previously reported in [5] for RCMs at 25 km resolution and Switzerland. A similar analysis to this study was performed for a sub region of the Alps (Po basin) with only one RCM (CCLM) and using MODIS snow cover [6]. This study indicates that their results scale to the whole Alpine domain, since the seasonal evolution of snow cover extent matches the one found here for CCLM (overestimation in winter, underestimation in spring; see Figure 3: inside Alpconv).

*3.3. Small-Scale Snow Cover Variability*

3.3.1. Isolated Effects of Altitude, Temperature, and Precipitation Differences

The small-scale snow cover biases were assessed by relating the differences in per grid cell snow cover to the differences in altitude, temperature and precipitation. The separate effects of each variable are shown in [22]. The strongest effect was for the altitude difference, such that snow cover was higher in RCMs compared to MODIS if the altitude in RCMs was also higher than in MODIS and vice versa. This effect was present, with different magnitudes, in all RCMs (both reanalysis and GCM driven), all months, and over all altitudes. The effect of temperature differences was not so strong, but still visible [22], such that snow cover in RCMs was higher compared to MODIS if temperature in RCMs was colder than in E-OBS and vice versa. The effect of precipitation was even lower than for temperature, but consistent in the sense that snow cover in RCMs was higher than in MODIS if precipitation was higher in RCMs compared to E-OBS and vice versa [22].

In the following the combined effect of each pairwise combination of altitude, temperature, and precipitation differences on snow cover biases is presented. Figure 4 shows the SNC bias in January in relationship to both differences in altitude and temperature for reanalysis driven RCMs (see Figure S7 for GCM driven RCMs and January, and [22] for all other months). Biases were low for the lowest and highest altitudes, because in winter these were either fully snow covered or snow free. For the middle altitudes (500–2000 m) there was a clear and strong relationship between the snow cover bias and the orography mismatch. The temperature bias had a smaller additional effect. SNC biases were lowest if both altitude and temperature differences were at their minimum.

Precipitation biases did not show any additional effect on snow cover biases in addition to the altitude differences, except for WRF331F, for which higher snow cover biases were linked to higher precipitation biases (see Figure S8 for January). However, their effect was much smaller than the effect of altitude differences. Figure S9 shows the effect of temperature and precipitation biases on snow cover biases in January. It confirms the previous two comparisons in that temperature biases have stronger effects than precipitation biases. However, it shows another pattern clearer than before: There were almost no observations that had at the same time low temperature and low precipitation biases, except for low altitudes up to 1000 m for CCLM4-8-17, RCA4, and WRF331F. Figures for all other months, as well as GCM driven RCMs can be found in [22].

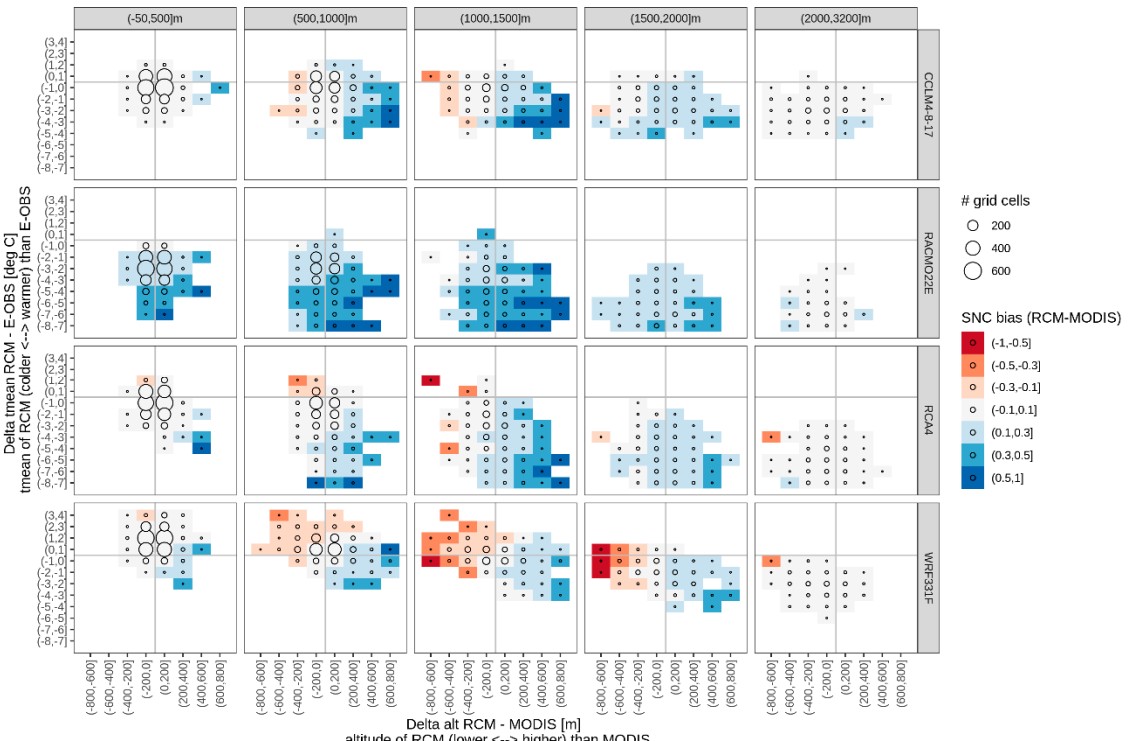

**Figure 4.** January snow cover bias in relation to altitude difference and temperature difference. The average snow cover fraction (SNC, unitless) bias (RCM–MODIS) in January is shown for classes of altitude difference (RCM–MODIS, x-axis) and classes of temperature difference (RCM–E-OBS). Rows are the different RCMs, and columns MODIS altitude classes. The area of the bubbles indicates the number of grid cells in the respective bin. SNC bias values were averaged over all grid cells in the respective bin, and the underlying grid cell data is averages for reanalysis driven RCMs and MODIS for the common period October 2002 to September 2008. Extreme values (delta temperature above 4 and below −8 degrees C, delta altitude above 800 or below −800 m) were excluded from this plot for better visualization and because their number is insignificant.

One reason that the temperature difference effect seemed lower than the altitude difference effect might be in the interactions, since the temperature difference was not corrected for the altitude difference between RCM and MODIS (only between RCM and E-OBS). So, the presented temperature differences can be regarded as independent of the altitude difference between RCM and MODIS. These temperature differences correspond to the cold biases that have been found in RCMs in the Alpine region [9]. We did not lapse rate adjust the temperatures for the difference between RCM and MODIS altitudes, because there was considerable variability in empirical monthly lapse rates in RCMs, which ranged from 4.8 to 9.1 °C/km (over all months and all RCMs), and differences between RCMs by month were up to 3.1 °C/km (see also Table S4).

### 3.3.2. Combined Effects of Altitude, Temperature, and Precipitation Differences

In order to quantify these effects, we employed simple linear regression models to derive the amount of explained variance by altitude difference, temperature difference, precipitation difference, and all three. We did not consider other explanatory variables, because our aim was not to create a good explanatory model, but only to quantify the effects of altitude, temperature and precipitation differences.

On average, the altitude difference could explain 19% of the variance in SNC bias, the temperature difference 11%, the precipitation difference 6%, and all together 31% (Table S5) for GCM driven RCMs. However, there were large differences between the months and altitude, and some also between RCMs. Most of the variance in snow cover bias could be explained in the months November to April and

for the middle altitudes (500–1500 m) with average R squared values of 52–65% depending on RCM (excluding HIRHAM5) with all differences as explanatory variables (Figure 5). In the higher altitudes, the explained variance dropped for the months December to February, because these areas were mostly snow covered in RCMs and MODIS, irrespective of potential biases. The maximum R squared could reach up to 73–87% depending on RCM, when looking at all altitudes and months.

The relative importance of the three explanatory variables for the total explained variance depended on altitude and RCM. For altitudes below 1000 m the temperature difference was the sole or more important variable for all RCMs. For altitudes above 1000 m the altitude difference was the sole or most important variable for ALADIN63, CCLM8-4-17, RCA4, and WRF331F. Temperature was an important variable for RACMO22E in general, and for WRF331F during spring. Precipitation was important only for WRF331F up to 1500 m, and for HIRHAM5 at highest altitudes >2000 m. The results did not differ substantially for reanalysis driven RCMs (Table S6 and Figure S10).

The low importance of precipitation can be caused by the questionable quality of E-OBS precipitation in the Alpine region. The station density differs strongly by nation and biases with respect to regional precipitation data sets with higher station density can reach up to RCM biases [23]. Possibly, better results could be obtained using a precipitation grid with a higher and more homogeneous station density for the Alps, such as e.g., [24]. However, neither E-OBS nor [24] are corrected for undercatch and uncertainties increase for small-scale features in all gridded precipitation data sets because of the more regional distribution of precipitation as compared to temperature [23].

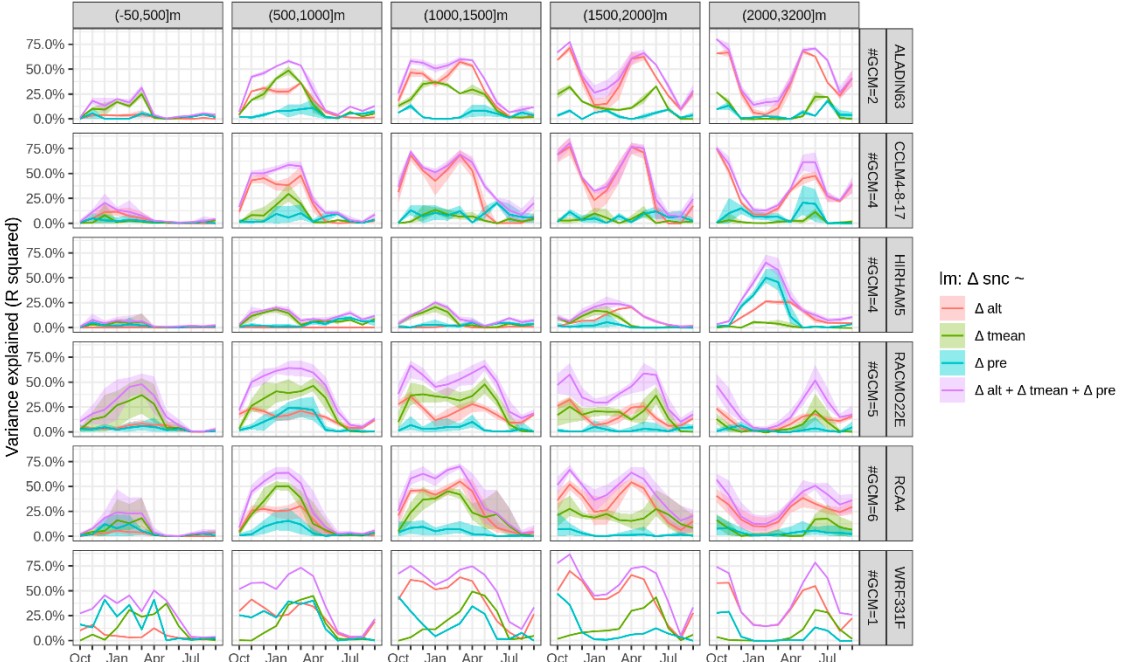

**Figure 5.** Explained variance in snow cover bias (RCM–MODIS) with linear regression models. R squared values (adjusted for the number of explanatory variables) were obtained from linear regression models of snow cover bias (RCM–MODIS) with different explanatory variables (covariates) (Δ alt: altitude difference (RCM–MODIS), Δ tmean: temperature difference (RCM–E-OBS), Δ pre: precipitation difference (RCM–E-OBS), Δ alt + Δ tmean + Δ pre: all three differences). Rows are the different RCMs, and columns MODIS altitude classes. RCMs are GCM driven; the number of different GCMs is given in row labels; the transparent band shows the minimum and maximum over all GCMs, while the solid line is the average.

### 3.3.3. Topography and Scale Considerations and the Added Value of Higher Resolutions

The small-scale biases in snow cover depended largely on the topography mismatch caused by the smoothing in the RCMs. Approximately half of the variability in snow cover across grid cells

in November to April could be explained by the differences between model and true topography. This is a crucial issue, since the importance of topography in complex mountain terrain is well known. However, RCMs encompass large regions, e.g., the European CORDEX domain covers all of Europe plus some surroundings, and in large parts of the domain the topography is well resolved, or its influence is negligible. Moreover, the benefits of higher resolutions in RCMs are debated (see e.g., the added value issue in [25]), also because the higher resolution EURO-CORDEX RCMs did not reduce biases in observed temperature and precipitation compared to previous lower resolution RCMs such as PRUDENCE [9], even in the complex Alpine terrain.

So, while for temperature and precipitation the added value of higher resolutions is generally unclear, we would argue that for precipitation and snow in complex terrain, higher resolutions have large added values. Using very-high resolution convection-permitting RCMs (~2 km horizontal grid spacing), clear benefits were found for precipitation in the Alps compared to coarser resolutions [2], however not for temperature. Similarly, snow amounts were better reproduced in these high resolution RCMs [3]. In another study concerning snow cover, biases were lowered by increasing the horizontal model resolution from 14 km to 8 km [6]. The findings in this study corroborate this issue by showing how much of the variability in snow cover was explained by topography mismatches.

These higher resolutions also open up the possibility to use RCMs outside their traditional use for temperature and precipitation studies, such as for snow. Many studies were performed since, covering different parts of the world, such as e.g., Spain [26], France [27], Northern Europe [28], the Colorado River Basin in the US [29], or Japan [30,31].

### 3.4. Comparison of Snow Depth in RCMs to Stations

Compared to observations from remote sensing, observations from meteorological stations were not homogenously distributed in space and especially not along the hypsometry of the studied region. Most of the stations lie in valleys, the number of stations declines with altitude and only very few exist at high elevations (e.g., >2000 m). This was also reflected in the distribution of the altitude differences between stations and model orography (Figures S11 and S12). For altitudes below 1000 m, the altitude difference was slightly skewed to positive differences (i.e., model orography higher than station elevation), but with most of the distribution centered around 0 m; the median difference was 76 m. For altitudes between 1000 m and 2000 m, most of the altitude differences were well above 0 m, with a median difference of 444 m. For altitude above 2000 m, only very few stations were available (up to 7), and some of them were located on mountain peaks, thus model orography was more than 1000 m below the station's elevation.

Figure 6 shows the relationship between difference in snow depth versus differences in altitude for January for reanalysis driven RCMs. As expected, there was again a clear relationship between biases in snow depth and orographic mismatch across all RCMs. Snow depth biases were lowest when altitude differences were lowest. This relationship was found for the other months as well and for GCM driven RCMs [22]. However, for the GCM driven runs, the RCMs had a positive bias (more snow) for altitudes >1000 m even when altitude differences were low, while for reanalysis driven RCMs, these biases centered around 0. This relationship was not present when looking at differences in temperature compared to differences in altitude [22]; only for the GCM driven runs, a minor correlation was observed.

The reason why no correlation was found for station data but for E-OBS might be two-fold. First, it could be related to scale issues of comparing points to grids. Second, adjusting all RCMs with the same lapse rate although they vary substantially (see also Table S4) might introduce spurious effects, which, combined with the actual temperature biases and local climate patterns, could results in arbitrary values. The altitude differences were higher for the stations than for the E-OBS grid, and thus the lapse rate correction had a stronger influence on the station data.

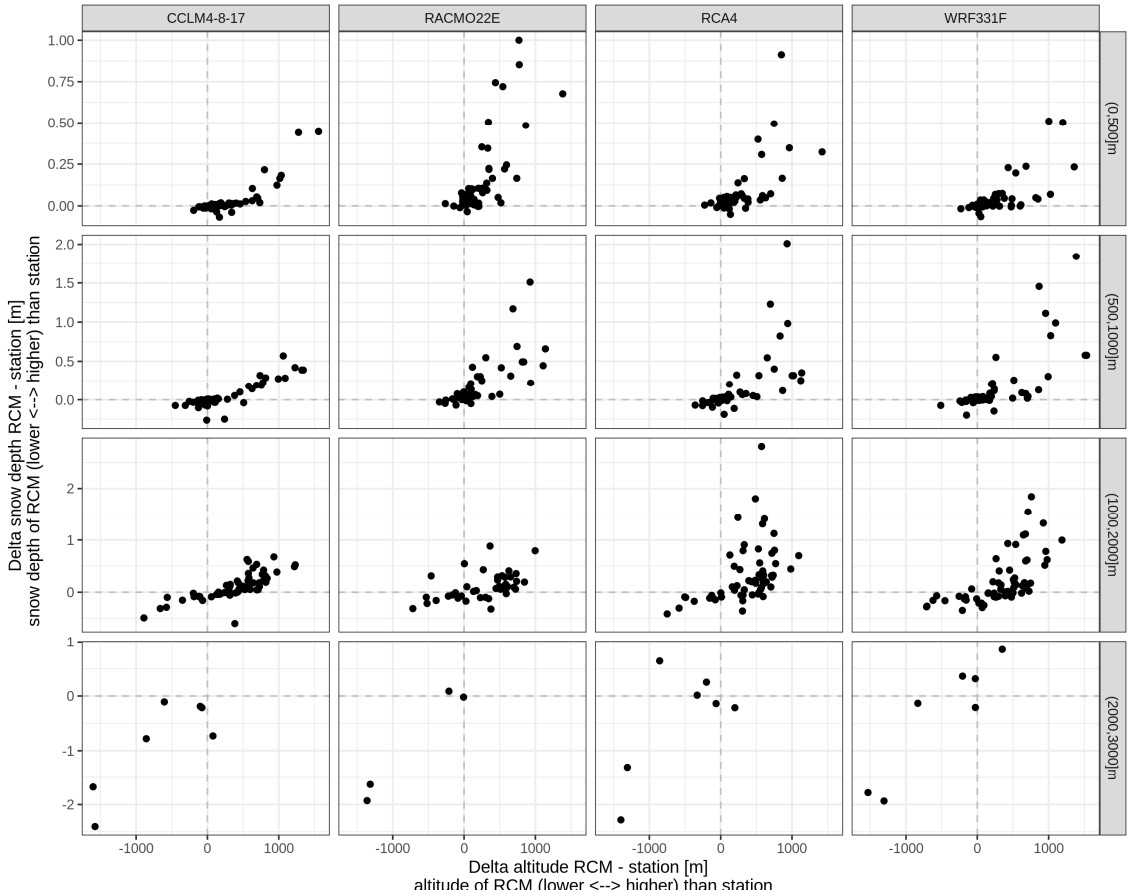

**Figure 6.** January snow depth bias (model–station observations) in reanalysis driven RCMs as function of altitude difference. The difference in snow depth is shown for the different RCMs (columns) and altitude classes of the stations (rows). Each point represents a station, and the difference in snow depth was calculated as the difference between the average monthly snow depth for all years that the station had in common with the RCM (during the period 1989–2008).

In order to quantify these biases and to compare the different RCMs, we calculated the median (and IQR; inter–quartile–range) snow depth bias for the stations that were within ±400 m of altitude difference with the RCMs. The ±400 m difference threshold was chosen in order to maximize the number of stations available, and at the same time reduce the influence of orography mismatch. Since each RCM orography differs, the station selection was performed separately per RCM. Average biases were then remarkably low (Table 2): Within −8 to +7 cm up to 2000 m of altitude for all reanalysis driven RCMs, except for RCA4 in spring months (March–May) between 1000 and 2000 m, where RCA4 overestimated snow depth on average by 19–34 cm.

Comparing the different RCMs with respect to the IQR (the bias range where half of the stations lay), CCLM4-8-17 had the lowest biases over all months and altitudes (largest IQR was −12 to 3 cm), followed by WRF331F, which slightly underestimated winter snow for altitudes between 1000 and 2000 m with IQRs between −8 to 6 cm and −25 to 1 cm. RACMO22E overestimated snow depth in December to March for altitudes below 1000 m with IQRs between 2 to 5 cm and 1 to 11 cm, and underestimated snow depth between 1000 m and 2000 m with IQRs up to −15 to 9 cm. RCA4 was comparable to CCLM4-8-17 and WRF331F for altitudes below 1000 m, however, between 1000 and 2000 m it had the largest IQRs of all RCMs going up to 14 to 108 cm. These large IQR of RCA4 might be related to the snow density parametrization, which can produce rather low snow densities in some cases (see also Figure S3).

**Table 2.** Average snow depth bias (RCM minus station) in cm for stations that are within ±400m of altitude difference with RCM orography. Values are median (IQR, inter–quartile–range, in parentheses) differences between four different reanalysis driven RCMs and stations, by elevation class of the stations, and month. The number of stations is not balanced between elevation classes, and is ~70 for (0, 500], ~46 for (500, 1000], and ~22 for (1000, 2000].

| Elevation Class [m] | Month | CCLM4-8-17 | RACMO22E | RCA4 | WRF331F |
|---|---|---|---|---|---|
| (0, 500] | Nov | 0 (0, 0) | 1 (0, 1) | 2 (1, 2) | 0 (0, 1) |
| (0, 500] | Dec | 0 (0, 0) | 3 (2, 5) | 3 (2, 4) | 1 (1, 2) |
| (0, 500] | Jan | 0 (−1, 0) | 3 (2, 7) | 3 (2, 4) | 0 (0, 1) |
| (0, 500] | Feb | 0 (−1, 0) | 4 (2, 9) | 3 (2, 4) | 0 (0, 1) |
| (0, 500] | Mar | 0 (0, 0) | 3 (1, 7) | 2 (2, 4) | 0 (0, 0) |
| (0, 500] | Apr | 0 (0, 0) | 0 (0, 1) | 1 (1, 1) | 0 (0, 0) |
| (0, 500] | May | 0 (0, 0) | 0 (0, 0) | 0 (0, 0) | 0 (0, 0) |
| (500, 1000] | Nov | −1 (−1, 0) | 1 (0, 1) | 1 (1, 2) | 1 (0, 1) |
| (500, 1000] | Dec | −1 (−3, 0) | 3 (1, 5) | 2 (0, 3) | 2 (0, 4) |
| (500, 1000] | Jan | −1 (−3, 0) | 4 (1, 7) | 1 (−1, 3) | 1 (−1, 4) |
| (500, 1000] | Feb | −3 (−6, −1) | 5 (1, 11) | 1 (−2, 3) | 0 (−2, 3) |
| (500, 1000] | Mar | −1 (−4, 0) | 5 (2, 10) | 2 (1, 4) | 0 (−1, 1) |
| (500, 1000] | Apr | 0 (0, 0) | 1 (0, 2) | 1 (1, 2) | 0 (0, 0) |
| (500, 1000] | May | 0 (0, 0) | 0 (0, 0) | 0 (0, 0) | 0 (0, 0) |
| (1000, 2000] | Nov | −1 (−2, 1) | 2 (0, 3) | 2 (0, 8) | 0 (−1, 3) |
| (1000, 2000] | Dec | −2 (−7, 0) | −3 (−4, 3) | −1 (−6, 22) | −5 (−8, 6) |
| (1000, 2000] | Jan | −3 (−8, 1) | −7 (−11, 3) | 1 (−11, 38) | −8 (−13, 6) |
| (1000, 2000] | Feb | −3 (−12, 3) | −8 (-15, 9) | 7 (−8, 57) | −7 (−17, 5) |
| (1000, 2000] | Mar | 1 (−7, 6) | −3 (-9, 17) | 19 (−1, 94) | −3 (−25, 1) |
| (1000, 2000] | Apr | −1 (−4, 1) | 6 (0, 33) | 24 (14, 108) | 0 (−6, 6) |
| (1000, 2000] | May | 0 (0, 0) | 1 (0, 19) | 34 (3, 72) | 0 (0, 1) |

When looking at GCM driven RCMs, the snow depth biases increased substantially compared to reanalysis driven RCMs (Figure 7). Median biases were no longer in the cm range, but up to 2 m for altitudes between 1000 and 2000 m. There were also more substantial differences between RCMs and the elevation classes, and also between GCMs. Similar to the reanalysis driven runs, CCLM4-8-17 had the lowest biases across all elevations, now in the range of −4 to 32 cm. For elevations below 1000 m, ALADIN63, HIRHAM5, and RCA4 were also comparable to CCLM4-8-17, however, with some over- and under-estimation. WRF331F overestimated snow depth below 1000 m, on average, up to 10 cm, and RACMO22E even more up to 19 cm. For altitudes between 1000 and 2000 m, average biases were lowest for CCLM4-8-17 and HIRHAM5, while for the other RCMs they reached values of ~1 m, and even >2 m for RACMO22E.

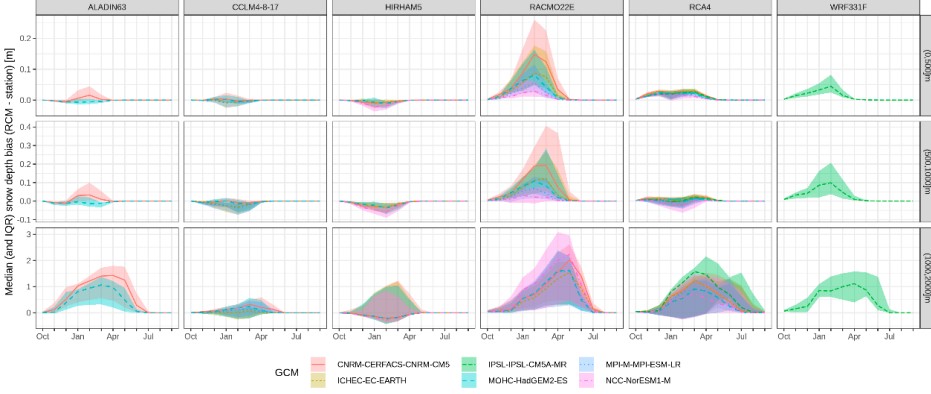

**Figure 7.** Average snow depth bias (RCM minus station) for stations that are within ±400 m of altitude difference with RCM orography. Lines indicate the median snow depth difference between the different RCMs and stations, and the transparent band is the IQR (inter–quartile–range). RCMs are shown in columns, with the driving GCM indicated in colors; rows are elevation classes of the stations. The number of stations is not balanced between elevation classes, and is ~55 for (0, 500], ~35 for (500, 1000], and ~9 for (1000, 2000].

### 3.5. Synthesis of the Evaluations for Snow Cover and Snow Depth

The evaluation of snow depth in RCMs was performed using large set of station observations covering the north, central and south-eastern parts of the Alps. Results should be taken cautiously, since point values do not represent area averages. However, results were in many aspects similar to other studies using gridded comparisons of snow amounts. For example, in this study snow depth biases in reanalysis driven RCMs were extremely low, but GCM driven RCMs had large positive biases. The same was found using gridded snow water equivalent products for the same RCMs and the same region (European Alps) [7]. Similarly, the overestimation of high-altitude snow amounts (and underestimation in low altitudes) reported in [5], was found also here, however, only for GCM driven RCMs in this study. For the reanalysis driven ones, in this study, snow depth biases were low across all altitudes (except for RCA4).

Snow cover extent is a sub-grid process in RCMs. It is typically parameterized as a function of snow amount. The parameterization differed strongly between RCMs, e.g., for CCLM8-4-17, snow cover and snow amount (or depth) were strongly linked, while for RCA4 the link was looser. Actually, for RCA4, time series of snow cover fraction of single pixels resembled much those of MODIS (not shown here), so the parameterization might even be adjusted empirically.

Because of this and the results of the evaluation of snow depth above, one cannot generalize the performance of each RCM in reproducing snow as a whole, but each snow variable has to be considered separately. A good representation of snow cover such as for RCA4 cannot guarantee similar results for snow depth, and vice versa: HIRHAM5 had snow depth estimates very close to observations, but its snow cover was substantially off.

Table 3 shows a summary of over- and under-estimation of snow cover and snow depth per RCM and month. While the elevation dependency of biases is not included in the table, it still gives an indication on the consistency or discrepancy in biases, such as e.g., the overestimation of both snow cover and snow depth for RACMO22E, or the underestimation of snow cover and at the same time overestimation of snow depth for GCM-driven HIRHAM5 and RCA4 in winter.

**Table 3.** Over- and under-estimation of snow cover fraction and snow depth in regional climate models. Indicative table summarizing average biases in snow cover fraction over all the study domain and snow depth over all stations (that is, ignoring any elevation dependency of biases). Meaning of cell contents from October to September: + (overestimation), − (underestimation), o (within tolerance level). First sign left of slash (/) stands for snow cover fraction and second sign for snow depth. The tolerance level is ±0.05 for snow cover fraction and ±5 cm for snow depth.

| RCM | GCM | Oct | Nov | Dec | Jan | Feb | Mar | Apr | May | Jun | Jul | Aug | Sep |
|---|---|---|---|---|---|---|---|---|---|---|---|---|---|
| | | | | *Reanalysis Driven* | | | | | | | | | |
| CCLM4-8-17 | ECMWF-ERAINT | o/o | o/o | o/o | o/o | o/o | −/+ | o/+ | o/o | o/o | o/o | o/o | o/o |
| RACMO22E | | o/o | o/o | +/+ | +/+ | +/+ | +/+ | +/+ | o/+ | o/o | o/o | o/o | o/o |
| RCA4 | | o/o | o/o | o/+ | o/+ | o/+ | o/+ | o/+ | o/+ | o/+ | o/+ | o/o | o/o |
| WRF331F | | o/o | o/o | o/+ | o/+ | o/+ | o/+ | o/+ | o/+ | o/o | o/o | o/o | o/o |
| | | | | *GCM Driven* | | | | | | | | | |
| ALADIN63 | CNRM-CERFACS-CNRM-CM5 | o/o | o/+ | o/+ | +/+ | +/+ | o/+ | +/+ | o/+ | o/+ | o/+ | o/o | o/o |
| | MOHC-HadGEM2-ES | o/o | o/+ | o/+ | o/+ | −/+ | o/+ | o/+ | o/+ | o/o | o/o | o/o | o/o |
| CCLM4-8-17 | CNRM-CERFACS-CNRM-CM5 | o/o | o/o | +/+ | +/+ | o/+ | o/+ | o/+ | o/o | o/o | o/o | o/o | o/o |
| | ICHEC-EC-EARTH | o/o | o/o | o/o | o/o | −/o | −/o | o/o | o/o | o/o | o/o | o/o | o/o |
| | MOHC-HadGEM2-ES | o/o | o/o | +/o | o/o | −/o | −/+ | o/+ | o/o | o/− | o/o | o/o | o/o |
| | MPI-M-MPI-ESM-LR | o/o | o/o | o/o | o/o | −/o | −/+ | o/+ | o/o | o/o | o/o | o/o | o/o |
| HIRHAM5 | CNRM-CERFACS-CNRM-CM5 | o/o | −/o | −/o | −/+ | −/+ | −/+ | −/+ | o/o | o/o | o/o | o/o | o/o |
| | ICHEC-EC-EARTH | o/o | −/o | −/o | −/+ | −/+ | −/+ | −/+ | o/o | o/o | o/o | o/o | o/o |
| | MOHC-HadGEM2-ES | o/o | −/o | −/o | −/+ | −/+ | −/+ | −/o | −/o | o/o | o/o | o/o | o/o |
| | NCC-NorESM1-M | o/o | −/o | −/o | −/o | −/+ | −/+ | −/o | −/o | o/o | o/o | o/o | o/o |
| RACMO22E | CNRM-CERFACS-CNRM-CM5 | o/o | +/+ | +/+ | +/+ | +/+ | +/+ | +/+ | +/+ | +/+ | o/o | o/o | o/o |
| | ICHEC-EC-EARTH | +/o | +/+ | +/+ | +/+ | +/+ | +/+ | +/+ | +/+ | o/+ | o/o | o/o | o/o |
| | MOHC-HadGEM2-ES | o/o | +/o | +/+ | +/+ | +/+ | +/+ | +/+ | +/+ | o/o | o/o | o/o | o/o |
| | MPI-M-MPI-ESM-LR | o/o | +/o | +/+ | +/+ | +/+ | +/+ | +/+ | +/+ | o/o | o/o | o/o | o/o |
| | NCC-NorESM1-M | o/o | +/o | +/+ | +/+ | o/+ | +/+ | +/+ | +/+ | o/+ | o/o | o/o | o/o |
| RCA4 | CNRM-CERFACS-CNRM-CM5 | o/o | o/+ | o/+ | +/+ | o/+ | o/+ | o/+ | o/+ | o/+ | o/+ | o/+ | o/o |
| | ICHEC-EC-EARTH | o/+ | o/+ | o/+ | o/+ | −/+ | o/+ | o/+ | o/+ | o/+ | o/+ | o/+ | o/+ |
| | IPSL-IPSL-CM5A-MR | o/+ | o/+ | o/+ | −/+ | −/+ | −/+ | o/+ | o/+ | o/+ | o/+ | o/+ | o/+ |
| | MOHC-HadGEM2-ES | o/o | o/o | o/o | o/+ | −/+ | −/+ | o/+ | o/+ | o/+ | o/+ | o/o | o/o |
| | MPI-M-MPI-ESM-LR | o/o | o/o | o/o | o/+ | −/+ | −/+ | o/+ | o/+ | o/+ | o/+ | o/+ | o/o |
| | NCC-NorESM1-M | o/o | o/o | o/+ | o/+ | −/+ | −/+ | o/+ | o/+ | o/+ | o/+ | o/+ | o/o |
| WRF331F | IPSL-IPSL-CM5A-MR | o/o | +/+ | o/+ | o/+ | o/+ | o/+ | +/+ | o/+ | o/+ | o/+ | o/o | o/o |

## 4. Conclusions

A two-fold evaluation of snow in the EURO-CORDEX regional climate models was performed using observations. First, snow cover was compared to remote sensing at the scale of the European Alps and at the grid cell level. Second, snow depth was compared to stations at the point level. The main conclusions are:

1.  RCMs were able to reproduce snow cover seasonality and amplitude at the scale of the European Alps fairly well, despite some over- and under-estimations depending on month and RCM.
2.  Reanalysis driven RCMs had lower biases than GCM driven RCMs for both snow cover and snow depth, implying that the forcing plays an important role.
3.  The orography mismatch, partly also the temperature and less the precipitation biases, exerted a strong influence on biases in snow variables. In regions with low altitude, temperature, and precipitation differences, RCMs showed minimal biases in snow cover and snow depth, implying that the snow schemes in the RCMs produce reasonable estimates with respect to their resolution.
4.  The parameterization of grid-scale snow cover fraction varies substantially amongst the RCMs. Because of this, RCMs that performed well for snow cover did not necessarily perform well for snow depth, and vice versa. Consequently, it is not possible to rank the RCMs in general terms for their capability to reproduce snow, as each variable (cover and depth) has to be considered separately.

The results of this evaluation give credibility to existing studies of snow projections using RCMs and allows climate modelers to further improve the representation of snow. Furthermore, it motivates the use of high-resolution RCMs for future snow studies in complex mountainous terrain.

**Supplementary Materials:** The following are available online at http://www.mdpi.com/2073-4433/11/1/46/s1, Figure S1: Grid cells excluded due to unrealistic snow accumulation; Figure S2: Distribution of station altitudes; Figure S3: Daily SNW versus SND for one year (2000) and all grid cells in the study region for individual RCMs; Figure S4: Daily SNC versus SNW for one year (2000) and all grid cells in the study region for individual RCMs; Figure S5: Comparison of GCM driven RCMs (historical period joined with RCP8.5) with observations (MODIS for snow cover, E-OBS for temperature and precipitation); Figure S6: Distribution of the differences between RCM and true (MODIS) altitude; Figure S7: January snow cover bias in relation to altitude difference and temperature difference; Figure S8. January snow cover bias in relation to altitude difference and precipitation difference; Figure S9. January snow cover bias in relation to temperature difference and precipitation difference; Figure S10: Explained variance in linear models of snow cover bias; Figure S11: Distribution of the differences between RCM and station altitude; Figure S12: Same as Figure S11, but for GCM driven RCMs; Table S1: Overview of applied regional climate model (RCM) runs; Table S2: Maximum daily snow depth (SND) and snow amount (SNW) simulated by the individual RCMs in the study region (historical and evaluation runs); Table S3: Number of total land grid cells in the study region and number of grid cells excluded due to unrealistic snow accumulation; Table S4: Empirical lapse rates in RCMs; Table S5: Percentage of variance in snow cover bias (RCM–MODIS) explained by differences in altitude, temperature, and precipitation with linear models; Table S6: Same as Table S5, but for reanalysis driven RCMs. More auxiliary material available at https://doi.org/10.5281/zenodo.3588775.

**Author Contributions:** Conceptualization, M.M., M.P., C.N., and M.Z.; methodology, M.M., C.N.; software, M.M.; validation, M.M.; formal analysis, M.M.; investigation, M.M. and M.P.; resources, M.M., M.P., C.N., and M.Z.; data curation, M.M., M.P., and C.N.; writing—original draft preparation, M.M.; writing—review and editing, M.M., M.P., C.N., and M.Z.; visualization, M.M.; supervision, M.P., C.N., and M.Z.; project administration, M.M.; funding acquisition, M.M., M.P., C.N., and M.Z. All authors have read and agreed to the published version of the manuscript.

**Funding:** This project has received funding from the European Union's Horizon 2020 research and innovation programme under the Marie Sklodowska-Curie grant agreement No 795310. The APC was funded by the European Union's Horizon 2020 research and innovation programme under the Marie Sklodowska-Curie grant agreement No 795310.

**Acknowledgments:** We acknowledge the World Climate Research Programme's Working Group on Regional Climate, and the Working Group on Coupled Modelling, former coordinating body of CORDEX and responsible panel for CMIP5. We also thank the climate modelling groups (listed in Table 1 of this paper) for producing and making available their model output. We also acknowledge the Earth System Grid Federation infrastructure an international effort led by the U.S. Department of Energy's Program for Climate Model Diagnosis and

Intercomparison, the European Network for Earth System Modelling and other partners in the Global Organisation for Earth System Science Portals (GO-ESSP). We acknowledge the E-OBS dataset from the EU-FP6 project UERRA (http://www.uerra.eu) and the Copernicus Climate Change Service, and the data providers in the ECA&D project (https://www.ecad.eu). We acknowledge the Hydrological office of Bolzano, Italy, the German Weather Service, and MeteoSWISS for the ground data. We thank Sven Kotlarski for discussions.

**Conflicts of Interest:** The authors declare no conflict of interest. The funders had no role in the design of the study; in the collection, analyses, or interpretation of data; in the writing of the manuscript, or in the decision to publish the results.

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
