# Peer review of "Evaluating Snow in EURO-CORDEX Regional Climate Models with Observations for the European Alps: Biases and Their Relationship to Orography, Temperature, and Precipitation Mismatches"

_atmosphere, doi:10.3390/atmos11010046_

Round 1

Reviewer 1 Report

A manuscript takes on a rarely discussed topic of snow biases in regional climate models for the very diverse orographically and climatically area of the Alps. It uses a large number of different data sources complemented each other. The selection of the data is well justified. All methodological procedures are perfectly explained (especially the procedure of cloud removal), and the interpretation of the results is clear and correct. It is well seen that the Authors put great effort into their research and preparation of the manuscript.

I recommend the manuscript to be published in journal Atmosphere. However, I would suggest Authors to consider following small comments:

line 24: I would suggest to add „Alps” and „mountain climate” to the Keywords; line 81: what does it mean „snow cover fraction”? The term should be explained here, where it is used firstly; move, please, here the explanation from the line 205; line 92: „SNW […] corresponds to SWE”: what is actually the difference between SNW and SWE? It should be explained; line 170: what was the altitude distribution of meteorological stations used in the study? line 208-209: LAEA, CDO – what does it mean? Lambert azimuthal equal-area? explain them, please; line 266-268: the information about station data has been already given in 170-172; it should be deleted here; line 301-302: „inner Alps”, „surroundings of the Alps”: what was the criterion for such a separation? line 323 and other similar sentences: „Shown are monthly averages…” it would be better to say „Monthly averages for […] are shown; line 378: are the R squared values corrected for the number of freedom degrees? line 419: „400 m” or „± 400 m”? line 424: „19-34” or „19-24” (according to the Table 2)? line 463: I suggest to add a table in Discussion with information of snow cover/ snow depth underestimated/ overestimated for each RCM in seasons; Figure S2: „SNW versus SNW” or „SNW versus SND”? correct it, please.

Reviewer 2 Report

See attached file for comments.
